# Nutritional Status of Slovene Adults in the Post-COVID-19 Epidemic Period

**Boštjan Jakše** [1,*] , **Uroš Godnov** [2] **and Stanislav Pinter** [3]

1    Independent Researcher, 1230 Domžale, Slovenia
2    Department of Computer Science, Faculty of Mathematics, Natural Sciences and Information Technologies, University of Primorska, 6000 Koper, Slovenia
3    Basics of Movements in Sport, Faculty of Sport, University of Ljubljana, 1000 Ljubljana, Slovenia
*    Correspondence: bj7899@student.uni-lj.si

**Abstract:** Background: Monitoring nutritional status data in the adult population is extremely important to mediate their health status. Unfortunately, for Slovenia (2.1 million European Union citizens), data on the body composition status of the general adult population are currently rare or nonexistent in scientific journals. Furthermore, dietary intake was last assessed several years before the COVID-19 epidemic period. Methods: We randomly recruited 844 adult Slovenes from all regions of Slovenia. The primary aim of the cross-sectional study was to examine body composition status (using a medically approved electrical bioimpedance monitor) during the post-COVID-19 epidemic period. In addition, we assessed dietary intake (using a standardized food frequency questionnaire) and compared the obesity propensity for both sexes separately using the body mass index (BMI) and body fat percentage (FAT%) obesity classification of the World Health Organization. Results: Regarding BMI classification, 43% of the whole sample was overweight (28%) or obese (15%), and there were more older adults than adults (64% vs. 42%, $p < 0.001$). The average FAT% of adult females and males was 26.9% and 19.5% ($p < 0.001$), respectively, while for older adult females and males, it was 32.7% and 23% ($p < 0.001$). In addition, a comparison of the proportions of obese people between the two cut-off obesity classifications (BMI vs. FAT%) showed a significantly underestimated proportion of obese female participants based on BMI classification (13% vs. 17%, $p = 0.005$). In terms of the dietary intake of the assessed nutrients in comparison with the national dietary reference values for energy and nutrient intake, the participants, on average, had lower intake than the recommended values for carbohydrates, fiber, vitamins C, D and E (for males) and calcium, and higher intake than the recommended values for total fat, saturated fatty acids, cholesterol, sodium and chloride (for males). Conclusions: The results urgently call for the need to not only improve the overall national nutritional status but also for regular national monitoring of body composition and dietary intake statuses.

**Keywords:** adults; post-COVID-19; nutritional status; body composition; body mass index; dietary intake; reference values

## 1. Introduction

The global increased incidence of being overweight and obese has been a major public health concern for several decades [1–3] and shows no sign of slowing down. Furthermore, a World Health Organization (WHO) report on the obesity epidemic in the European region showed that 60% of adults are either overweight or obese [3]. In fact, the obesity rate continues to increase substantially globally and is associated with a constant rise in medical and economic costs [4]. Analysis of the prevalence of adult obesity in 20 European countries showed that Slovenia is the country with the highest prevalence of obesity (20.8%) [5]. By 2025, in the WHO European region, the obesity rate is projected to increase in 44 countries [6]. This is important in terms of the adverse health dimensions of the

obesity epidemic, which is associated with over 20 major diseases and over 50 obesity-related comorbidities that mediate this global burden and manifest in several ways [7,8]. Additionally, obesity itself accelerates the normal aging process, and age acceleration is linked to a higher risk of premature mortality [9].

Monitoring the prevalence of being overweight and obese is of great importance for assessing dietary and lifestyle interventions aimed at preventing or reducing the burden of obesity. Body mass index (BMI) is typically used to define the proportion of overweight and obese people in epidemiological studies. However, the BMI classification has, on many occasions, been criticized for its lack of sensitivity in distinguishing between fat mass and lean mass for any given BMI [10,11]. Therefore, simply relying on BMI to estimate the prevalence of being overweight and the obesity rate could hinder governmental dietary and lifestyle interventions aimed at obesity prevention and control.

Importantly, the quality of BMI data across Europe is oftentimes questionable in terms of the quality and frequency of BMI measurements at the national level [6]. Nevertheless, for Slovenia, the latest scientifically published data on BMI were measured as a part of a national dietary study in 2017/2018. The average BMI of adults ($n$ = 364) and older adults ($n$ = 416) who were overweight was 26.7 kg/m$^2$ and 28.4 kg/m$^2$, respectively, and altogether, 59% of adults and 74% of older adults were overweight or obese [12]. The adult data for Serbia (a country that is in the same region but is not included in the European region) from the National Health Survey in 2013 showed similar results (e.g., 60.5% of adults were overweight or obese) [13]. Furthermore, in line with these results, in one of our previous studies of 151 adult Slovenes who had been on a plant-based diet for varying lengths of time, we also measured the subjects' baseline body composition (BC) data before the diet change. The average baseline BMI of the sample was in the overweight category (26.4 kg/m$^2$); however, 50% of subjects were overweight or obese [14].

Due to the limitations of BMI, in our study, we combined it with BC measurements, which may be important and useful for smaller-scale observational studies and for people with sarcopenic obesity [15]. Therefore, the aim of this study was to assess the nutritional status (i.e., BC and dietary intake status) of Slovene adults (i.e., adults and older adults) in the post-COVID-19 epidemic period. In fact, measuring the BC of the adult Slovenian population is particularly important, since there is not yet such a comprehensive measurement of Slovenian adults published in scientific journals. Additionally, using a large sample of adults with measured BC status, we compared the prevalence of obesity using two obesity classification cut-offs (BMI and body fat percentage (FAT%)).

## 2. Materials and Methods

### 2.1. Study Design and Eligibility

This cross-sectional study protocol was reviewed and approved on 26 June 2022 by the Ethical Committee in the field of sports in Slovenia (approval document no. 033–41/2022–5), and the trial was registered on 30 June 2022, at https://clinicaltrials.gov with number NCT05438966. Furthermore, the study was conducted from 1 July to 31 August 2022 and evaluated various locations in Slovenia to cover all regions [16].

To randomly recruit the participants, we contacted numerous representatives from hotels, municipalities, and local fire stations whose premises are used around the country (e.g., conference rooms, meeting rooms, municipal and firemen's premises and booklets). In addition, we also used social media groups to identify locations, dates and types of various free programs (except for registration fees) that were taking place for citizens (i.e., seminars, workshops, trainings, lectures and courses). Furthermore, for the organizers of the various programs, we offered a free BC analysis with interpretation and printouts sent to the individual's e-mail; in return, we obtained and anonymously used the measured data. In addition, we randomly sent a standardized food frequency questionnaire (FFQ) to females and males via e-mail. All included participants were given the complete explanation of the study and provided their written informed consent. Importantly, the participants were not remunerated financially.

*2.2. Subjects*

To obtain as much representative insight into the actual nutritional status of the average adult citizens in the post-COVID-19 epidemic period (which was a study aim), we conducted the study using various free social programs that were not intentionally or specifically related to a healthy and active lifestyle (i.e., physical activity and nutrition). Therefore, in the recruitment process, we randomly invited healthy adults (≥18 years) without BMI restrictions to participate; top-level or competitive athletes, pregnant or lactating women and adults with active common chronic diseases (i.e., the criterion was the current use of medication for any common chronic disease) were not included in the study.

In the final analysis, we enrolled 844 Slovene adults who underwent the BC measurement. In addition, we randomly sent a standardized FFQ to subjects whose BC was measured (more precisely, to 60 females and 60 males), and 39 fully completed questionnaires were returned (22 from females and 17 from males). A detailed flow chart of the research recruitment process is presented in Figure 1.

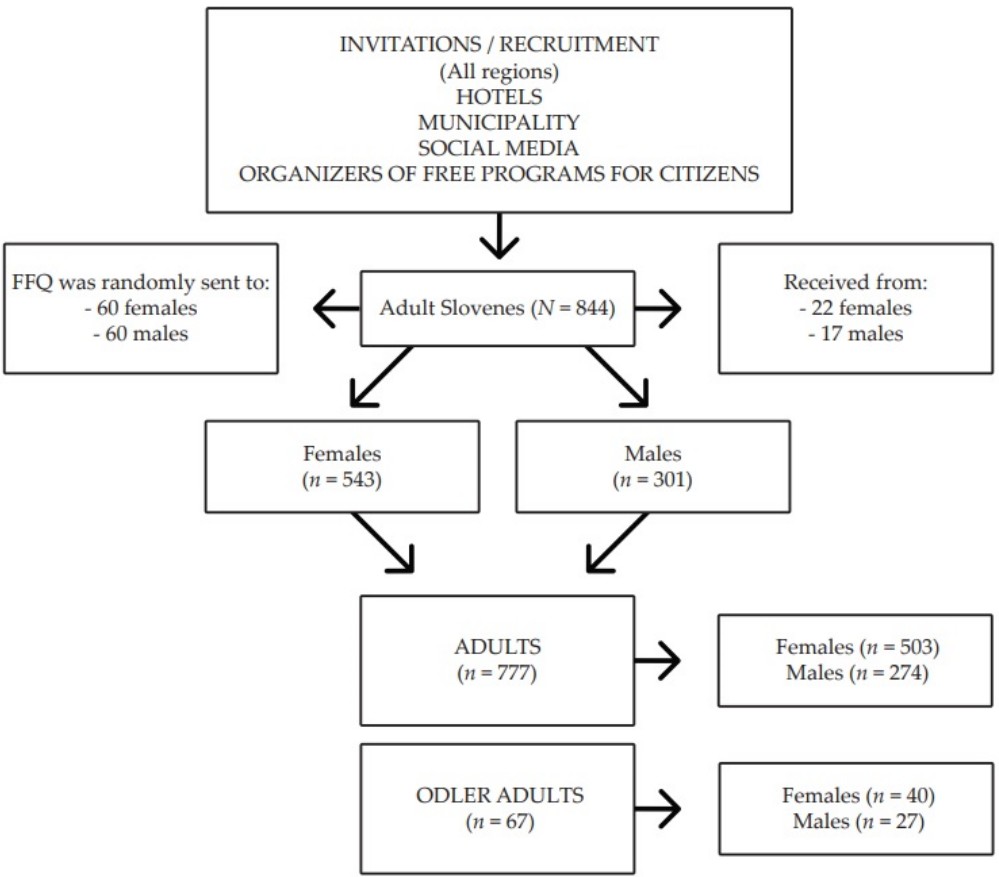

**Figure 1.** Recruitment process for participation in the study.

*2.3. Outcomes*

2.3.1. Body Composition Measures

The BC measures were obtained with a medically approved and calibrated electrical bioimpedance monitor (Tanita 780 S MA, Tokyo, Japan). Body height (BH) was measured with a standardized, medically approved professional personal floor scale with a stand (Kern, MPE 250K100HM, Kern & Sohn, Balingen, Germany). In addition, BMI was calculated as body mass (BM) in kilograms, measured by the BC monitor, and was divided by the square of BH in meters. The BC results included the following variables: BH, BM, BMI, FAT in % and kg, FATM/BH (kg/cm$^2$), fat-free mass (FFM) in % and kg, FFM/BH (kg/cm$^2$), total body water (TBW) and phase angle (PhA).

Furthermore, we compared the obtained results of the BC indices with the recommended targets for BMI and FAT% obesity classification [17,18]. In addition, we compared the proportions of obese people between these two classifications for the category of obesity for both sexes and the whole sample (i.e., BMI for obesity class (i.e., ≥30 kg/m² for females and males) [17] vs. FAT% for obesity class (i.e., >35% for females and >25% for males)) [18].

### 2.3.2. Dietary Intake

To estimate the participants' dietary intake, we used a standardized FFQ [19,20]. The FFQ was translated from the original Dutch language into the Slovenian language by a professional translator and has already been used in Slovenia in several studies [21–23]. Foods and ingredients extracted from the FFQ were carefully entered into the national Open Platform for Clinical Nutrition (OPEN) [24,25], which is a national web-based application developed by the Jožef Stefan Institute in Slovenia and supported by the European Federation of the Association of Dietitians [24]. Importantly, the vast majority of subjects did not consume any dietary supplements; therefore, we did not include dietary supplements in the final analysis.

Energy and nutrient intake were compared with Slovenian national dietary reference values [26] that are based on Central European reference values (German (D), Austrian (A) and Swiss (CH) (D-A-CH) [27,28]). The current Slovenian national dietary reference values do not mention the reference values for saturated fatty acids (SFAs), monounsaturated fatty acids (MUFAs), polyunsaturated fatty acids (PUFAs), cholesterol, free sugar or biotin intake; therefore, the intake of these nutrients (except for free sugar) was compared with the D-A-CH reference [27,28], while the intake of free sugar was compared with the UK Scientific Advisory Committee on Nutrition (SACN) recommendations (<5% of daily energy intake) [29].

### 2.4. Statistical Analysis

Statistical analysis was performed using R 4.1.1. with the tidyverse [30], ggstatsplot [31] and arsenal [32] packages. The tidyverse package was used for data transformation, ggstatsplot2 was used for data visualization and arsenal was used for statistical calculation. For numerical variables, we used independent sample *t*-tests, and for non-parametric variables, we used the Mann–Whitney U test. To determine the differences in the obesity proportion between the two obesity classification methods (BMI and FAT%), we used the z-test for proportion equality with continuity correction. The threshold for statistical significance was $p < 0.05$. There were no missing data. Data are presented as the means (standard deviations).

## 3. Results

### 3.1. Characteristics of the Participants

The whole sample included 844 Slovene adults (543 females (64%) and 301 males (36%), $p = 0.304$). Within the sample, there were 777 adults (18–64 years) vs. 67 older adults (≥65 years) ($p < 0.001$). The average ages of females and males were $41 \pm 13$ and $40 \pm 14$ years, respectively ($p = 0.304$). In addition, the average age of adults and older adults was $38.3 \pm 11$ and $68 \pm 4$ years, respectively ($p < 0.001$). Furthermore, the sample represented the country in a relatively balanced regional manner (e.g., 15% from North/Northwest, 38% from Central, 30% from South/Southwest and 17% from East/Northeast Slovenia). Most of the sample were married or in a partnership (64%), lived outside the city in suburban and rural environments (77%) and were employed or self-employed (81%). Furthermore, the minority of the whole sample had a university education or above (43%), and only 13% were older adults. Regarding smoking status, 25% of the whole sample reported currently smoking. To identify the individual dietary pattern, in the FFQ, we had four options for choosing a dietary pattern (i.e., vegan (no animal-origin food intake), vegetarian (no meat or fish intake), sometimes vegetarian (several times

weekly) and omnivores), and the majority of participants were omnivores (61%), while the remaining participants were vegetarians and sometimes vegetarians (no vegans).

### 3.2. Body Composition Status

For the whole sample, the average BMI status was in the overweight BMI category. In terms of sex differences, females had significantly lower BMI values, higher FAT% and FATM/BH values and lower FFM% and FFM/BH values. The average FAT% for adult females and males were 26.9% and 19.5%, respectively, while for older adult females and males, it was 32.7% and 23%, respectively. Importantly, adults were, on average, within (i.e., borderline) the normal BMI category, while older adults were in the overweight BMI category and had higher FAT% and lower FFM% values. Furthermore, adult females and males had significantly higher FFM% values than older adult females and males ($73.1 \pm 7.6\%$ vs. $67.3 \pm 6.5\%$, $p < 0.001$ and $80.5 \pm 7.0\%$ vs. $77.2 \pm 7.4\%$, $p = 0.028$). The complete BC status is shown in Table 1.

**Table 1.** Body composition status by sex and age.

| Parameter | BH (cm) | BM (kg) | BMI (kg/m²) | FAT (%) | FATM (kg) | FATM/BH (kg/cm) | FFM (%) | FFM (kg) | FFM/BH (kg/cm) | TBW (kg) | PhA (°) |
|---|---|---|---|---|---|---|---|---|---|---|---|
| Total | 170.9 ± 9.0 | 73.6 ± 17.2 | 25.1 ± 5.0 | 24.6 ± 8.3 | 18.7 ± 9.7 | 0.11 ± 0.06 | 75.4 ± 8.3 | 55.0 ± 11.6 | 0.32 ± 0.05 | 39.2 ± 8.3 | 5.9 ± 0.7 |
| Females | 166.3 ± 6.3 | 67.2 ± 13.9 | 24.3 ± 4.8 | 27.3 ± 7.6 | 19.2 ± 9.3 | 0.12 ± 0.06 | 72.7 ± 7.8 | 48.0 ± 6.1 | 0.29 ± 0.03 | 34.3 ± 4.4 | 5.6 ± 0.5 |
| Males | 179.1 ± 7.0 | 85.3 ± 16.4 | 26.6 ± 5.0 | 19.8 ± 7.1 | 17.8 ± 10.3 | 0.10 ± 0.06 | 80.1 ± 7.1 | 67.5 ± 8.2 | 0.38 ± 0.04 | 48.1 ± 6.1 | 6.4 ± 0.7 |
| *p*-value | **<0.001** | **<0.001** | **<0.001** | **<0.001** | 0.057 | **<0.001** | **<0.001** | **<0.001** | **<0.001** | **<0.001** | **<0.001** |
| Adults | 171.2 ± 8.9 | 73.4 ± 17.4 | 24.9 ± 5.0 | 24.3 ± 8.2 | 18.4 ± 9.7 | 0.11 ± 0.06 | 75.7 ± 8.2 | 55.0 ± 11.6 | 0.32 ± 0.05 | 39.3 ± 8.3 | 6.0 ± 0.7 |
| Older adults | 167.3 ± 8.5 | 76.5 ± 14.6 | 27.2 ± 4.2 | 28.7 ± 8.4 | 22.2 ± 8.2 | 0.13 ± 0.05 | 71.3 ± 8.4 | 54.3 ± 11.4 | 0.32 ± 0.72 | 38.1 ± 7.9 | 5.3 ± 0.6 |
| *p*-value | **<0.001** | 0.162 | **<0.001** | **<0.001** | **0.002** | **<0.001** | **<0.001** | 0.631 | 0.717 | 0.252 | **<0.001** |
| Adults Females | 166.6 ± 6.2 | 67.0 ± 14.1 | 24.1 ± 4.9 | 26.9 ± 7.6 | 18.8 ± 9.3 | 0.11 ± 0.06 | 73.1 ± 7.6 | 48.2 ± 6.1 | 0.29 ± 0.03 | 34.5 ± 4.4 | 5.7 ± 0.5 |
| Males | 179.5 ± 7.1 | 85.1 ± 16.8 | 26.4 ± 5.0 | 19.5 ± 7.0 | 17.6 ± 10.4 | 0.10 ± 0.06 | 80.5 ± 7.0 | 67.6 ± 8.4 | 0.38 ± 0.04 | 48.3 ± 6.2 | 6.6 ± 0.6 |
| *p*-value | **<0.001** | **<0.001** | **<0.001** | **<0.001** | 0.077 | **<0.001** | **<0.001** | **<0.001** | **<0.001** | **<0.001** | **<0.001** |
| Older adults Females | 161.7 ± 5.0 | 69.1 ± 11.0 | 26.4 ± 4.2 | 32.7 ± 6.5 | 23.1 ± 7.6 | 0.14 ± 0.05 | 67.3 ± 6.5 | 45.9 ± 4.7 | 0.28 ± 0.03 | 32.4 ± 3.4 | 5.1 ± 0.5 |
| Males | 175.6 ± 5.0 | 87.4 ± 12.3 | 28.4 ± 4.1 | 22.8 ± 7.3 | 20.7 ± 8.9 | 0.12 ± 0.05 | 77.2 ± 7.4 | 66.7 ± 5.2 | 0.38 ± 0.02 | 46.6 ± 4.0 | 5.6 ± 0.7 |
| *p*-value | **<0.001** | **<0.001** | 0.063 | **<0.001** | 0.261 | **0.044** | **<0.001** | **<0.001** | **<0.001** | **<0.001** | **<0.001** |

Data are the means ± standard deviations (SDs). Statistically significant values are shown in bold. A *t*-test was used. BH: body height, BM: body mass, BMI: body mass index, FAT: body fat, FATM: body fat mass, FFM: fat-free mass, TBW: total body water, PhA: whole body phase angle.

Analysis of the proportions of normal-weight participants by sex showed that significantly more females than males (64% vs. 45%, $p < 0.001$) and more adults than older adults (58% vs. 36%, $p < 0.001$) were within the normal BMI category. Importantly, the age pattern within the BMI classification was seen among both sexes (e.g., the youngest participants were in the normal category). For example, the average age for females in the normal BMI category was higher for both overweight and obesity class 1 and lower for obesity class 3 (e.g., females: $38.7 \pm 12.7$, $44.2 \pm 12.9$, $50.7 \pm 12.8$, $41.9 \pm 13.5$ and $42.5 \pm 7.3$ years ($p < 0.001$); males: $35.3 \pm 12.4$, $42.5 \pm 14.0$, $47.4 \pm 13.0$, $40.2 \pm 14.3$ and $44 \pm 10.4$ years ($p < 0.001$)).

In addition, the comparison of the proportions of obese subjects between the two obesity classifications (i.e., BMI vs. FAT%) showed a significant difference for the total sample (16% vs. 20%, $p = 0.002$) and for the female group (13% vs. 17%, $p = 0.005$). However, for the male group, the difference in the proportions of obese subjects for both classifications was not significantly different (20% vs. 23%, $p = 0.171$). The direction of the difference (underestimation or overestimation) in both obesity classifications for females may depend on the average BMI and average FAT% data for overweight females, which was 29.3 kg/m² and $32.7 \pm 3.9\%$, respectively. The complete BMI and FAT% obesity classifications are presented in Table 2.

**Table 2.** Body mass index and FAT% obesity classification.

| Parameter | Total | Females | Males | Adults | Females | Males | Older Adults | Females | Males |
|---|---|---|---|---|---|---|---|---|---|
| According to BMI classification (*n*/%) | | | | | | | | | |
| Normal (BMI 18.5–24.9 kg/m$^2$) | 479 (57) | 345 (64) | 134 (45) | 455 (58) | 327 (65) | 128 (47) | 24 (36) | 18 (45) | 6 (22) |
| Overweight (BMI 25–29.9 kg/m$^2$) | 233 (28) | 127 (23) | 106 (35) | 209 (27) | 115 (23) | 94 (34) | 24 (36) | 12 (30) | 12 (44) |
| Obesity 1 class (BMI 30–34.9 kg/m$^2$) | 86 (10) | 40 (7) | 46 (15) | 69 (9) | 31 (6) | 38 (14) | 17 (25) | 9 (23) | 8 (30) |
| Obesity 2 class (BMI 35–39.9 kg/m$^2$) | 18 (2) | 10 (2) | 8 (3) | 16 (2) | 9 (2) | 7 (3) | 2 (3) | 1 (3) | 1 (4) |
| Obesity 3 class (BMI > 40 kg/m$^2$) | 28 (3) | 21 (4) | 7 (2) | 28 (4) | 21 (4) | 7 (3) | 0 | 0 | 0 |
| *p*-value | | | | | **<0.001** | | | | |
| According to FAT% obesity classification | | | | | | | | | |
| Female >35% | | 92 (17) | | | 77 (15) | | | 15 (38) | |
| Female <35% | | 451 (83) | | | 426 (85) | | | 25 (62) | |
| Male >25% | | | 75 (25) | | | 64 (23) | | | 11 (41) |
| Male <25% | | | 226 (75) | | | 210 (77) | | | 16 (59) |
| *p*-value | | **0.005** | | | **0.005** | | | 0.789 | |

Statistically significant values are shown in bold. Body mass index (BMI) and FAT% obesity classifications by the WHO [17,18].

### 3.3. Dietary Intake Status

Tables 3 and 4 show the intake of energy, macro- and micronutrients (13 vitamins, 6 minerals and 3 trace elements) of Slovene adults. The estimated nutrient intake, compared with the dietary reference values, showed a lower intake of carbohydrates (recommended >50% E) and fiber (recommended ≥30 g/d) and a higher intake of total fat (recommended 30% E) and SFAs for males (recommended ≤10% E) and cholesterol (recommended <300 mg/d). Importantly, total protein (and also animal protein intake) (per g and in % E) was the only macronutrient that was significantly higher for males compared with females.

**Table 3.** Intake of energy and macronutrients.

| Macronutrients (per Day) | Females (*n* = 22) | Males (*n* = 17) | Total (*n* = 39) | *p*-Value |
|---|---|---|---|---|
| Energy intake (kcal) | 1950 ± 142 | 2097 ± 399 | 2014 ± 421 | 0.213 |
| Carbohydrates (g) | 216 ± 61 | 198 ± 64 | 208 ± 64 | 0.428 |
| (% E) | 44 ± 6 | 38 ± 11 | 42 ± 10 | **0.034** |
| Total sugars$^{TS}$ (g) | 70 ± 57 | 56 ± 25 | 64 ± 46 | 0.702 |
| Free sugars$^{FS}$ (g) | 23 ± 37 | 10 ± 11 | 17 ± 29 | 0.340 |
| (% E) | 4 ± 6 | 2 ± 2 | 3 ± 5 | 0.186 |
| Starches (g) | 60 ± 38 | 79 ± 40 | 68 ± 40 | 0.106 |
| Dietary fibers (g) | 27 ± 6 | 26 ± 7 | 26 ± 6 | 0.619 |
| (% E) | 3 ± 1 | 3 ± 1 | 3 ± 1 | 0.321 |
| Fat (g) | 79 ± 31 | 91 ± 30 | 84 ± 31 | 0.240 |
| (% E) | 36 ± 10 | 39 ± 11 | 37 ± 10 | 0.213 |
| SFAs (g) | 22 ± 13 | 25 ± 11 | 23 ± 12 | 0.161 |
| (% E) | 10 ± 4 | 11 ± 5 | 10 ± 4 | 0.336 |
| MUFAs (g) | 27 ± 15 | 27 ± 15 | 27 ± 15 | 0.898 |
| (% E) | 12 ± 6 | 12 ± 7 | 12 ± 6 | 0.488 |
| PUFAs (g) | 19 ± 6 | 18 ± 7 | 18 ± 6 | 0.436 |
| (% E) | 9 ± 3 | 8 ± 2 | 8 ± 2 | 0.157 |
| Cholesterol (mg) | 307 ± 276 | 511 ± 352 | 395 ± 323 | 0.079 |
| Proteins (g) | 80 ± 22 | 108 ± 32 | 92 ± 30 | **0.007** |
| (% E) | 17 ± 4 | 21 ± 5 | 18 ± 5 | **0.012** |
| Plant proteins (g) | 39 ± 12 | 34 ± 15 | 37 ± 13 | 0.234 |
| (% E) | 8 ± 3 | 6 ± 3 | 7 ± 3 | **0.023** |

**Table 3.** *Cont.*

| Macronutrients (per Day) | Females (*n* = 22) | Males (*n* = 17) | Total (*n* = 39) | *p*-Value |
|---|---|---|---|---|
| Animal proteins (g) | 41 ± 19 | 74 ± 35 | 55 ± 31 | **0.004** |
| (% E) | 9 ± 3 | 14 ± 6 | 11 ± 6 | **0.003** |
| Total water$^{TW}$ (L) | 1.7 ± 0.5 | 2.1 ± 0.5 | 1.9 ± 0.5 | **0.013** |

Data are the means ± standard deviations (SDs). Significant differences in values are shown in bold. The nonparametric Mann–Whitney U test was used. % E = percentage of total energy intake (general Atwater energy conversion factors were used (kcal/g): carbohydrates and proteins = 4, dietary fibers = 2, fat = 9, alcohol = 7 [33]. TS = total sugars: all monosaccharides and disaccharides: free sugars plus sugars that are naturally present in foods (e.g., lactose in milk, fructose in fruits) [34]. FS = free sugar: all monosaccharides and disaccharides that are added to foods and beverages by the manufacturer, cook or consumer (i.e., added sugars) plus sugars that are naturally present in honey, syrups, fruit juices, fruit juice concentrates and sports drinks (defined by the WHO [34] and adapted by the SACN [29]). SFAs = saturated fatty acids; MUFAs = monounsaturated fatty acids; PUFAs = polyunsaturated fatty acids. TW = total water: from beverages and solid foods.

**Table 4.** Intake of selected vitamins, minerals and trace elements.

| Micronutrients (per Day) | Females (*n* = 22) | Males (*n* = 17) | Total (*n* = 39) | *p*-Value |
|---|---|---|---|---|
| **Vitamins** | | | | |
| Thiamine (mg) | 1.4 ± 0.4 | 1.7 ± 0.6 | 1.6 ± 0.5 | 0.100 |
| Riboflavin (mg) | 1.5 ± 0.6 | 2.2 ± 0.9 | 1.8 ± 0.8 | **0.008** |
| Niacin (mg) | 16 ± 5 | 25 ± 13 | 20 ± 10 | **0.005** |
| Pantothenic acid (mg) | 5.7 ± 2.2 | 7.3 ± 2.7 | 6.4 ± 2.5 | **0.041** |
| Vitamin B6 (mg) | 1.7 ± 0.6 | 1.9 ± 0.6 | 1.8 ± 0.6 | 0.371 |
| Biotin (µg) | 60 ± 31 | 78 ± 35 | 68 ± 34 | 0.081 |
| Folate (µg) | 378 ± 152 | 420 ± 176 | 396 ± 162 | 0.420 |
| Vitamin B$_{12}$ (µg) | 4.0 ± 3.4 | 6.1 ± 4.2 | 4.9 ± 3.9 | 0.084 |
| Retinol equ.$^{RE}$ (mg) | 2.1 ± 0.4 | 2.2 ± 0.2 | 2.1 ± 0.3 | 0.103 |
| Vitamin C (mg) | 78 ± 55 | 61 ± 33 | 71 ± 47 | 0.411 |
| Vitamin D (µg) | 3.8 ± 2.8 | 5.3 ± 3.5 | 4.5 ± 3.2 | 0.145 |
| Vitamin E (mg) | 13 ± 3 | 12 ± 5 | 13 ± 6 | 0.661 |
| Vitamin K (µg) | 205 ± 139 | 158 ± 126 | 185 ± 134 | 0.077 |
| **Minerals** | | | | |
| Calcium (mg) | 725 ± 344 | 891 ± 366 | 798 ± 359 | 0.208 |
| Magnesium (mg) | 496 ± 149 | 470 ± 140 | 485 ± 144 | 0.661 |
| Phosphorus (mg) | 1624 ± 460 | 1800 ± 415 | 1701 ± 445 | 0.213 |
| Potassium (mg) | 3513 ± 1064 | 3172 ± 866 | 3364 ± 985 | 0.343 |
| Sodium (mg) † | 1604 ± 1303 | 1755 ± 934 | 1670 ± 1145 | 0.357 |
| Chloride (mg) † | 2131 ± 951 | 2684 ± 1446 | 2372 ± 1207 | 0.208 |
| **Trace elements** | | | | |
| Iron (mg) | 17 ± 5 | 17 ± 4 | 17 ± 4 | 0.831 |
| Zinc (mg) | 11 ± 3 | 11 ± 3 | 11 ± 4 | 0.372 |
| Selenium (µg) | 71 ± 34 | 107 ± 60 | 87 ± 50 | 0.052 |

Data are the means ± standard deviations (SDs). Significant differences in value are shown in bold. The nonparametric Mann–Whitney U test was used. RE = retinol equivalent: vitamin A + α-carotene (1 mg retinol equivalent = 12 mg α-carotene) + β-carotene (1 mg retinol equivalent = 6 mg β-carotene) + γ-carotene (1 mg retinol equivalent = 12 mg γ-carotene). † Sodium and chloride intake are from food only (i.e., without salt from meal preparation).

Insufficiencies in micronutrient intake were found for vitamins C (recommended intake of 95 mg/d and 110 mg/d for females and males), D (recommended intake of 20 µg/d), E (for males) (recommended intake of 13–15 mg/d) and calcium (recommended intake of 1000 mg/d), while sodium and chloride (for males) intake exceeded the recommended intake. Niacin, riboflavin and pantothenic acid were the only micronutrients for which the intake was significantly different between females and males.

## 4. Discussion

### 4.1. Main Findings

The present study aimed to investigate the nutritional status of Slovene adults in the post-COVID-19 epidemic period, with the primary outcomes related to their BC status. Analysis of a large sample of adults showed that a high proportion of adults and older adults were in the overweight and obese BMI categories, with more males than females. However, females had significantly higher FAT% and lower FFM% values. In addition, the proportion of obese subjects based on BMI and FAT% obesity classifications showed a lower proportion of obese females in the BMI obesity category, with the theoretical possibility of underestimation or overestimation using only the BMI obesity classification tool. Furthermore, the estimated dietary intake of Slovene adults compared with the dietary reference values showed an unbalanced dietary pattern, namely, a lower (complex) carbohydrate intake (i.e., whole grains and legumes) and a higher intake of highly processed and/or nutrient-depleted foods (i.e., white bread and pasta, vegetable oils and butter). Specifically, we estimated a lower intake of macronutrients than the recommended intake for carbohydrates and fiber and within the recommendation of free sugar intake, while simultaneously estimating a higher intake of total fat, SFAs (for males) and cholesterol. In terms of micronutrient adequacy, we found a lower intake of vitamins C, D and E (for males) and calcium but a higher intake of sodium and chloride (for males) than the recommended intake.

### 4.2. Body Composition Status

In our study, we found that 42% of adults and 64% of older adults were either in the overweight or obese BMI category. In addition, the average FAT% for adult females and males were 26.9% and 19.5%, respectively, while for older adult females and males, they were 32.7% and 23%, respectively. Our results are consistent with those from before the COVID-19 pandemic period in Slovenia in terms of the pattern of results obtained, although the proportion of subjects in the overweight and obese categories and the FAT% value in our study were lower for both sexes and age groups. Finally, the BMI and FAT% obesity classification comparison suggests that the female obesity rate with a cut-off of >35% for FAT% may be underestimated, at least in our study, when the average BMI and FAT% values for females in the overweight category were 29.3 kg/m$^2$ and 32.7%, respectively.

According to a recent Slovenian nationally representative dietary survey (SI.Menu 2017/2018) of 780 adults and older adults, as many as 59% and 74% of adults and older adults were in the overweight or obese BMI category [12]. Unfortunately, there are no other recent past or current (post-COVID-19 epidemic period) data on obesity status in Slovenian adults. Nevertheless, we have data from one large Slovenian study (*n* = 8036) of adolescents from 2014 that showed a similar trend, namely, 29% of females and 38% of males in this study were overweight or obese [35]. In addition, some data from the SiMenu 2017/2018 study were published only in a scientific monograph [36] and may be used for comparison with our results, which is perhaps relevant. The researchers used electrical bioimpedance (Tanita BC 730) for BC analysis and found that the average FAT% for adult females and males was 33% and 24.9%, respectively, while that for older adult females and males was 37.5% and 29%, respectively [36]. Of note, we were not able to find data on the sample size on which the BC measurements were performed. However, in our previous study of 151 people with plant-based diets, we found that their baseline BC status (before they went on a plant-based diet) was in line with both mentioned results [14]. Specifically, 50% of the subjects were overweight or obese, while their baseline FAT% was 28.7%. The data are of importance since the study sample also covers all Slovenian regions and had significantly more females than males, similar to our study.

It Is well established that obesity is defined as the accumulation of excess body fat and not simply an excess of BM. This fact is very important, for a significant proportion of individuals in the overweight BMI category have an increased FFM% and low FAT% or are within the normal BMI category with low FFM% and increased FAT% (i.e., sarcopenic

obesity) [15,37]. It is becoming clear that BMI is a rather poor indicator of FAT% (and it does not capture fat segmentation) [11,38]; in addition, we are seeing an increased rate of sarcopenic obesity that is currently also becoming a major public health challenge, with an estimated worldwide prevalence of up to 42% of adults [39,40]. That being said, BC provides important prognostic information on an individual's BM management and mortality risk that is not provided by traditional proxies of adiposity, such as BMI [37,41,42].

### 4.3. Dietary Intake Status

The estimated average dietary intake indicates the presence of an unbalanced diet. The most obvious deviations from the dietary reference values were seen for carbohydrate, fiber, total fat and SFAs, cholesterol, vitamins C, D and E (for males) and calcium intake. Interestingly, males consumed significantly more total protein and animal protein than females ($41 \pm 19$ (9% of E) vs. $74 \pm 35$ g/d (14% E), $p = 0.004$ ($p = 0.003$)).

A higher intake of animal protein, especially for males, is logical in the annual meat consumption statistics for Slovene adults (calculated from the age of zero and including discarded meat). The average Slovene adult eats as much as six times more than the sustainable average annual amount of meat, set in the framework of the Planetary diet (calculated from the age of two onwards) [43]. Furthermore, the average weekly intake of meat and meat products of Slovenian adults were estimated to be four times higher than the reference values of the Planetary diet [36,44].

Compared with our results, the abovementioned Slovenian nationally representative dietary survey (SI. Menu 2017/2018) revealed greater nutritional insufficiency for the nutrients for which the researchers have already published data, for example, for fiber (20.9–22.4 g/d vs. 26 g/d), vitamin D (2.45–2.8 µg/d vs. 4.5 µg/d) and folate intake (294.6–295.5 µg/d vs. 396 µg/d) but not for vitamin B$_{12}$ (5–6.2 µg/d vs. 4.9 µg/d) [12,45–47] and free sugar intake (6.4–6.5% E vs. 3% E) [48]. Although it is clear from the available data that the adequate nutrient intake of a Slovenian adult is problematic, we emphasize that the method of assessing the energy and nutrient intake of the Si.Menu 2017/2018 study was more rigorous (i.e., the dietary intake data were collected with two nonconsecutive 24 h dietary recalls complemented with a food propensity questionnaire) [12,45–48]. Although there are few current data on the nutritional intake of the Slovenian adult population, these trends are consistent in several ways with an older, first national study on the dietary intake of Slovenian adolescents ($n = 1813$) aged 14–17 years, for example, in excess SFA and sodium intake and insufficient vitamin D and calcium intake [49]. In addition, the results for the adolescents differed from the results of both studies for adults in terms of excess free sugar intake, adequate fiber intake and insufficient folate and PUFA intake. All these available data indicate the less-than-optimal dietary habits of Slovenians. Importantly, the results of both studies of adults estimated that the unbalanced diet is tied to the dietary reference values for a mixed (omnivorous) diet, with most adults eating this way. However, in the future, there will likely be an actual need to differentiate dietary reference values for other increasingly recognized dietary patterns (e.g., Mediterranean, low-fat vegan or low-carbohydrate, high-fat (ketogenic) diets) that show various more or less (un)favorable health outcomes.

Finally, the researchers evaluated various health benefits (e.g., BM loss, effect on cardiovascular/type 2 diabetes risk factors and disease outcomes) and potential risks or limitations of recently trendy dietary patterns (i.e., low-carbohydrate, high-fat (ketogenic) diet [50–56], Mediterranean diet [57,58], vegetarian diet [56,59–65] and omnivorous diet [21,66]). The weight of evidence strongly supports a balanced (well-designed) diet and simultaneously allows for various variations and interpretations of research results. Although further research is always needed, a comparison of different dietary patterns has shown that a well-designed dietary pattern should primarily be based on unprocessed or minimally processed plant-based food [44,67]. The mechanisms of action are known and described in great detail elsewhere [60,68–75]. Regardless, it is necessary to be aware that a sustainable diet can only be built on a multidisciplinary approach, as due to various objec-

tive challenges/limitations in society (e.g., sociological, physiological, ethnic-traditional, geographical and economic), an easy shift to a stricter plant-based dietary pattern, therefore, cannot be a "one-way street" [76]. Nevertheless, the beginning of changes for the better in terms of improving global human health and environmental issues can be the implementation of many sustainable dietary patterns [77], which would lead to a reduction in the currently excessive production and intake of meat [78] and a greater intake of healthy plant-based foods [79], with which we can have a beneficial effect on the health of the environment and reduce the risk of common chronic non-communicable diseases [67,77].

## 5. Strengths and Limitations

This is the first study in Slovenia that examined the BC status of Slovene adults and the first post-COVID-19 epidemic study of adults that assessed their nutritional status (i.e., BC and dietary intake). Our study has several strengths and limitations that are worth further consideration. Notably, the strength of our study is that it was performed in the post-COVID-19 epidemic period; therefore, it presents insight into the current status of the population, as well as the possible consequences of the two-year complete and partial lockdowns due to the pandemic. In the study, we randomly included all measured adults within the inclusion criteria and with relatively balanced regional coverage. The studied sample was large in terms of the number of adult inhabitants of Slovenia and the largest to date to measure BC. In addition, we also randomly assessed the dietary intake of the studied sample, which can be considered a strength.

However, we acknowledge a few important limitations. Our study did not examine the physical activity status of the sample or its possible impact on BC and dietary intake status. Furthermore, the recruitment method probably left out some representatives of groups that may not be as socially active; consequently, we did not know their nutritional status and its possible effect on the obtained results. In addition, this group of adults may belong to a lower socioeconomic class. Studies have shown that socioeconomic factors contribute to obesity at the individual and community levels [4]. However, a recent systematic review of 21 studies suggested the simultaneous existence of both social causality (e.g., effect of income on obesity) and reverse social causality (e.g., effect of obesity on income) [80]. Therefore, in this regard, a generalized simplification, even at the community level, may not be appropriated. Finally, for dietary intake, we used an FFQ and not several-day-weighted dietary records, which are more accurate. In addition, we sent the FFQ to random subjects who were also included in the BC measurement rather than to all subjects. However, the FFQ method was given to a large sample, and the strategy used without external financial resources was our compromised decision. In addition, in the final analysis, we did not include the nutrient intake from dietary supplements; therefore, we do not know the state of intake of dietary supplements nor their impact on the overall dietary intake status. However, we encourage authorities in Slovenia to establish standardized and continuous monitoring of the nutritional status of adults on a national level.

## 6. Conclusions

To the best of our knowledge, this is the first study published in scientific journals that assessed the BC status of Slovene adults. The findings from our study showed that a large proportion of adults (42%) and older adults (64%) were in either the overweight or obese categories. Furthermore, the BMI and FAT% obesity classification comparison probably underestimated the proportion of females with obesity when using only the BMI obesity classification tool. Furthermore, the lack of data on the BC status of Slovene adults calls for regular monitoring of the prevalence of being obese and overweight as well as dietary intake. In addition, the dietary intake of adults suggests that adults' diets differ significantly and unfavorably from the dietary reference values for balanced omnivorous diets, which is already noticeable as a consequence of the burden on public health.

**Author Contributions:** Conceptualization, B.J. and S.P.; methodology, U.G., B.J. and U.G.; software, U.G.; formal analysis, U.G.; investigation, B.J., S.P. and U.G.; writing—original draft preparation, B.J.; writing—review and editing, B.J., U.G. and S.P.; visualization, B.J. and U.G.; supervision, U.G. and S.P. All authors have read and agreed to the published version of the manuscript.

**Funding:** This research received no external funding.

**Institutional Review Board Statement:** This study protocol was reviewed and approved on 26 June 2022 by the Ethical Committee in the field of sports in Slovenia (approval document no. 033–41/2022–5), and the trial was registered on 30 June 2022, at https://clinicaltrials.gov with number NCT05438966.

**Informed Consent Statement:** Written informed consent has been obtained from the participants to publish this paper.

**Data Availability Statement:** The data used to support the findings of this study are included within the article.

**Acknowledgments:** The authors wish to thank all participants for their collaboration in our study. This work would not have been possible without them.

**Conflicts of Interest:** The authors declare no conflict of interest related to this manuscript.

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
