# Peer review of "Nutritional Status of Slovene Adults in the Post-COVID-19 Epidemic Period"

_ejihpe, doi:10.3390/ejihpe12120122_

Round 1

Reviewer 1 Report

I gave all recommendations in the attached file.

Reviewer 2 Report

Dear authors,

congratulations for your valuable study. The study underlines the urgent need for obesity and weight management. However, my advice is to include more references about the relationship between overweight and the health status of individuals. Here are some very good articles: 

Kahleova, H.; Fleeman, R.; Hlozkova, A.; Holubkov, R.; Barnard, N.D. A plant-based diet in overweight individuals in a 16-week randomized clinical trial: Metabolic benefits of plant protein. Nutr. Diabetes 2018, 8, 1–10.

Kahleova, H.; Dort, S.; Holubkov, R.; Barnard, N.D. A Plant-Based High-Carbohydrate, Low-Fat Diet in Overweight Individuals in a 16-Week Randomized Clinical Trial: The Role of Carbohydrates. Nutrients 2018, 10, 1302.

Kahleova, H.; Hlozkova, A.; Fleeman, R.; Fletcher, K.; Holubkov, R.; Barnard, N.D. Fat Quantity and Quality, as Part of a Low-Fat, Vegan Diet, Are Associated with Changes in Body Composition, Insulin Resistance, and Insulin Secretion. A 16-Week
Randomized Controlled Trial. Nutrients 2019, 11, 615.

Kahleova, H.; Rembert, E.; Alwarith, J.; Yonas, W.N.; Tura, A.; Holubkov, R.; Agnello, M.; Chutkan, R.; Barnard, N.D. Effects of a Low-Fat Vegan Diet on Gut Microbiota in Overweight Individuals and Relationships with Body Weight, Body Composition, and Insulin Sensitivity. A Randomized Clinical Trial. Nutrients 2020, 12, 2917.

Mishra, S.; Xu, J.; Agarwal, U.; Gonzales, J.R.; Levin, S.A.; Barnard, N.D. A multicenter randomized controlled trial of a plant- based nutrition program to reduce body weight and cardiovascular risk in the corporate setting: The GEICO study. Eur. J. Clin. Nutr. 2013, 67, 718–724.

Chainani-Wu, N.; Weidner, G.; Purnell, D.M.; Frenda, S.; Merritt-Worden, T.; Pischke, C.; Campo, R.; Kemp, C.; Kersh, E.S.; Ornish, D. Changes in Emerging Cardiac Biomarkers after an Intensive Lifestyle Intervention. Am. J. Cardiol. 2011, 108, 498–507. 
